# Urinary Metabolomics around Parturition Identifies Metabolite Alterations in Dairy Cows Affected Postpartum by Lameness: Preliminary Study

**Guanshi Zhang [1,†,‡]**, **Elda Dervishi [1]**, **Grzegorz Zwierzchowski [1,§]**, **Rupasri Mandal [2]**,
**David S. Wishart [2]** and **Burim N. Ametaj [1,*]**

[1]  Department of Agricultural, Food and Nutritional Science, University of Alberta, Edmonton, AB T6G 2P5,
    Canada; guanshi@ualberta.ca (G.Z.); dervishi@ualberta.ca (E.D.);
    grzegorz.zwierzchowski@uwm.edu.pl (G.Z.)
[2]  Departments of Biological Sciences and Computing Science, University of Alberta, Edmonton, AB T6G 2E9,
    Canada; mandal@ualberta.ca (R.M.); dwishart@ualberta.ca (D.S.W.)
[*]  Correspondence: bametaj@ualberta.ca; Tel.: +1-780-492-9841
[†]  Present address: Center for Renal Precision Medicine, Division of Nephrology, Department of Medicine,
    The University of Texas Health San Antonio, San Antonio, TX 78229, USA.
[‡]  Present address: Audie L. Murphy Memorial VA Hospital, South Texas Veterans Health Care System,
    San Antonio, TX 78229, USA.
[§]  Present address: Faculty of Biology and Biotechnology, University of Warmia and Mazury,
    1a Oczapowskiego str., Olsztyn, 10-719, Poland.

**Abstract:** (1) Background: The objective of this study was to evaluate the urine of dairy cows for presence of metabolites with the potential to be used as screening biomarkers for lameness as well as to characterize pre-lame, lame, and post-lame cows from the metabolic prospective. (2) Methods: Six lame and 20 control healthy cows were used in this nested case-control study. Urinary 1H-NMR analysis was used to identify and measure metabolites at five time points including −8 and −4 weeks prepartum, lameness diagnosis week (1–3 weeks postpartum) as well as at +4 and +8 weeks after calving. (3) Results: A total of 90 metabolites were identified and measured in the urine. At −8 and −4 weeks, 27 prepartum metabolites were identified as altered, at both timepoints, with 19 and 5 metabolites excreted at a lower concentration, respectively. Additionally, a total of 8 and 22 metabolites were found at greater concentration in pre-lame cows at −8 and −4 weeks, respectively. Lame cows were identified to excrete, at lower concentrations, seven metabolites during a lameness event with the top five most important metabolites being Tyr, adipate, glycerate, 3-hydroxy-3-methylglutarate, and uracil. Alterations in urinary metabolites also were present at +4 and +8 weeks after calving with N-acetylaspartate, glutamine, imidazole, pantothenate, beta-alanine and trimethylamine, with the greatest VIP (variable importance in projection) score at +4 weeks; and hipurate, pantothenate 1,3-dihydroxyacetone, galactose, and Tyr, with the greatest VIP score at +8 weeks postpartum. (4) Conclusions: Overall, results showed that urine metabotyping can be used to identify cows at risk of lameness and to better characterize lameness from the metabolic prospective. However, caution should be taken in interpretation of the data presented because of the low number of replicates.

**Keywords:** urine; metabolomics; dairy cows; biomarkers

## 1. Introduction

Lameness is related to hoof and leg inflammatory conditions that are associated with pain that impairs normal walking or posture [1]. Almost 90% of lameness is identified in the foot area, with most

cases (>80%) occurring in the hind limbs [2]. Lameness has a multifactorial etiology and it is commonly classified into two main types: (1) non-infectious and (2) infectious [3,4]. Non-infectious lameness includes laminitis, white line disease, sole ulcers, and joint and upper leg trauma and deformity, whereas infectious lameness includes heel warts, digital and interdigital dermatitis, and foot rot. Some of the most important etiological factors of lameness include diet, breed, environment, and genetics [5]. Diets containing high proportions of grain, rich in starch, increase the odds of lameness, especially during early lactation when cows transition from low-grain diet prepartum to a high-grain diet after calving [6]. Environmental factors such as hard flooring, stall type, stress related to bullying of lower rank animals and manure are some additional causal agents [7]. Finally, genetic factors also influence lameness, including the higher susceptibility of Holstein cows and hardness of the hoof [8].

Lameness is a very costly pathology, costing dairy producers around 469 USD per case [9]. Given that the incidence of lameness averages from 23% to 50% or higher, this makes lameness one of the costliest diseases of dairy cattle. The main negative effects of lameness are related to the delayed resumption of ovarian activity [10] where lame cows, with high score of 4, have a 3.5 times greater likelihood of having a delayed ovarian cyclicity compared with the normal cows. Melendez et al. [11] found that cows becoming lame within the first 30 days of lactation have lower conception rates (17.5 percent versus 42.6 percent), lower overall pregnancy rate (85 percent versus 92.6 percent) and a higher incidence of cystic ovarian disease (25 percent versus 11 percent). Lameness also has been shown to have a negative impact on milk yield. We reported a decrease of 31% in milk yield in Holstein dairy cows during the week of lameness diagnosis and an overall decrease of 13% during the first 56 d in milk [12]. Also, Hernandez et al. [13] showed that lameness decreased milk yield by 7%.

Currently, there are no urine tests available to identify cows at risk of lameness. Moreover, not much is known with regards to urine composition prior to, during, and after lameness diagnosis. Metabolite fingerprinting of urine prior to, during, and after lameness occurrence can give insights into the pathogenesis as well as the potential hidden causal agents of lameness in dairy cows. Screening urinary samples through metabolomics for metabolite fingerprints is very important because: (1) it can identify cows at risk of lameness at a very early stage when the disease is at subclinical stages, (2) it can contribute to better understanding the pathobiology of lameness, (3) it can help to better characterize lameness from the metabolic prospective; (4) it can provide knowledge about disease progression, and (5) it can help to monitor the efficiency of a therapeutic intervention or potential complications.

Additionally, using urine to test the susceptibility of dairy cows to lameness or other periparturient diseases has important advantages because the collection of urine is non-invasive, easy, and less expensive. Previously we reported that pre-lame, lame, and post-lame cows had alterations in pro-inflammatory cytokines (interleukin-(IL)-6) and acute phase proteins [serum amyloid A (SAA) and haptoglobin (Hp) prior to and during lameness diagnosis and that those cows were in a state of metabolic acidosis, with blood lactate higher in pre-lame and lame cows during −8 and −4 weeks prepartum as well as during lameness diagnosis week compared with their healthy counterparts (2–3 weeks postpartum) [12]. We hypothesized that pre-lame, lame, and post-lame cows might have significant alterations in metabolites in the urine that can be used for developing screening tests for risk of lameness as well as for better understanding the disease progression and recovery after medical treatment. Therefore, the objective of this study was to analyze urinary samples with proton nuclear magnetic resonance ($^1$H-NMR) in order to identify potential alterations in the urine at pre-clinical stages of lameness, during full clinical stage of the lameness, and after medical treatment of dairy cows for lameness.

## 2. Materials and Methods

The current study was part of a larger project designed to identify the risk biomarkers of several periparturient diseases in dairy cows. All experimental procedures were approved by the University of Alberta Animal Policy and Welfare Committee for Livestock (Approval #AUP00000129), and animals were cared for in accordance with the guidelines of the Canadian Council on Animal Care [14].

The metabolomics analyses were performed at the Metabolomics Innovation Centre, University of Alberta, Edmonton, AB, Canada.

*2.1. Animals and Monitoring of Clinical Health Status*

One hundred pregnant Holstein dairy cows at the Dairy Research and Technology Centre, University of Alberta (Edmonton, AB, Canada), were used in a nested case-control study. Details of the animals and diets are described by Zhang et al [12]. Briefly, the experimental period lasted for 16 w, starting from −8 weeks before parturition to +8 weeks postpartum for each cow. Twenty healthy cows with no clinical disease during the entire experimental period were selected to be analyzed at −8, −4 and at one week of lameness. At weeks +4 and +8 postpartum, 20 healthy cows (Controls - CON) were selected. Six pregnant multiparous Holstein dairy cows (parity: 3.0 ± 0.6, Mean ± SEM) were diagnosed with lameness (diagnosed between +2 and +3 weeks postpartum). Healthy CON cows were similar in parity (3.3 ± 0.6), and body condition score (BCS for Con vs. Lame group, 2.87 vs. 2,62, respectively). If a cow with lameness was diagnosed as having other diseases it was excluded from the analysis. Healthy cows had no clinical signs of any disease including metritis, lameness, milk fever, mastitis, retained placenta, or ketosis. Locomotion scoring of cows for diagnosis of lameness was based on a protocol developed by Sprecher et al. (1997). This locomotion scoring assesses cows with a 5-point locomotion scoring system (1 = normal walking and 5 = severely lame). Briefly, locomotion score 1 is assigned to normal cows (cow walks and stand normally); locomotion score 2 is given to cows that are mildly lame (cow that stands flat and arches when walks); locomotion score 3 is ascribed to cows that are moderately lame (cow stands and walk with arched back and short strides); locomotion score 4 is given to cows that are lame (cow walks and stands with arched back, favoring one limb); locomotion score 5 describes cows that are severely lame (pronounced arching of back and reluctant to move). Diets were offered as TMR for ad libitum intake once daily at 0800 h to allow approximately 5% orts. All total mixed rations (TMR) were formulated to meet or exceed the nutrient requirements of dry and early lactating dairy cows with an estimated body weight of 680 kg as per National Research Council guidelines [15].

The health status of all cows was monitored daily, starting at −8 weeks prior to the expected date of calving and continuing up to +8 weeks postpartum. Diagnosis of lameness was based on a locomotion procedure described by Sprecher et al. [16]. The locomotion score of the cows diagnosed with lameness ranged between 3–5. Lame cows displayed signs of lameness, including an arched-back posture and a gait of one step at a time. Lame cows favored one or more of their limbs/feet. Sprecher et al. [16] locomotion score involves a 5-point scale system with a locomotion score of 1 (LCS 1) assigned to cows with normal gait and an LCS of 5 to cows with severe lameness. The 20 healthy cows were ranked with LCS 1 (normal gait). Cows affected by lameness were treated by a trimming specialist and medicated if needed. Treatment information for lame cows was reported previously [12].

*2.2. Urine Sample Collection*

Urine samples were obtained from 100 transition Holstein dairy cows once per week at 0700 before feeding from −8 weeks before parturition to +8 weeks postpartum. A mid-stream sample of naturally voided urine sample was collected by gently massaging the perineal area in a sterile tube of 20 mL. Fecal material or other debris from the exterior of the vulva were removed by washing the area with warm water and soap and the area was disinfected with alcohol. Twenty healthy CON cows and six cows diagnosed only with lameness were selected for further metabolomics analyses. $^1$H-NMR analyses were conducted on urine samples from five timepoints: at −8 (53–59 days) and −4 weeks (25–31 days) before parturition, the disease week and at +4 (25–31 days) and +8 weeks (53–59 days) after calving from each cow. Urine samples were stored at −80 °C until analysis to avoid loss of bioactivity and contamination. All samples were thawed on ice for approximately 2 h before use.

### 2.3. NMR Compound Identification and Quantification

Sample preparation procedures were described in Dervishi et al. [17]. Briefly, urine samples were centrifuged for 5 min at 10,000 rpm (Eppendorf Centrifuge 5424, Eppendorf AG, Hamburg, Germany) at +4 °C. Afterwards, a 600 μL aliquot of sample supernatant was added in an Eppendorf tube (1.5 mL) followed by the addition of 70 μL of $D_2O$ (Deuterium oxide, Sigma-Aldrich Co., St. Louis, MO, USA) and 30 μL of a standard phosphate buffer solution [11.667 mM DSS (disodium-2,2-dimethyl-2-silapentane-5-sulphonate), 730 mM imidazole, and 0.47% $NaN_3$ in $H_2O$]. Urine samples were vortexed, mixed and then centrifuged (Heraeus Instruments GmbH, Germany) under 10,000 rmp × 5 min × 4 °C. The urine samples (700 μL) were then transferred to a standard glass NMR tube (5 mm thin wall, Wilmad LabGlass, Vineland, NJ, USA) for subsequent NMR spectral analysis. The pH of urine samples ranged from 7.3 to 7.7.

All proton $^1$H-NMR spectra were obtained on a 500 MHz Inova spectrometer (Varian Inc., Palo Alto, CA, USA) equipped with a 5 mm hydrogen, carbon, and nitrogen (HCN) Z-gradient pulsed-field gradient (PFG) Varian cold-probe. $^1$H-NMR spectra were acquired at 25 °C using the first transient of the Varian tnnoesy-presaturation pulse sequence, which was chosen for its high degree of quantitative accuracy [18]. Spectra were collected with 128 transients and eight steady-state scans using a 4 s acquisition time and a 1 s recycle delay [19].

The $^1$H-NMR spectra were processed and analyzed with the Chenomx NMR Suite Professional software package (version 7.6, Chenomx Inc., Edmonton, AB, Canada) as previously described [20,21]. Prior to spectral analysis, all free induction decays (FIDs) were zero-filled to 64,000 datapoints and line-broadened 0.5 Hz. The singlet produced by DSS was used as an internal standard for both chemical shift-referencing (set to 0 ppm) and metabolite quantification. Each spectrum was processed and analyzed independently by at least two experienced NMR spectroscopists to minimize errors in compound identification and quantification.

### 2.4. Statistical Analysis

All metabolite concentrations measured were normalized to each urine sample's corresponding creatinine value (assuming a constant rate creatinine excretion for each urine sample) to compensate for variations in urine volume. The concentration of each metabolite is expressed as μM/mM creatinine. Univariate analysis of data was performed using Wilcoxon–Mann–Whitney (rank sum) test provided by R [22]. Statistical significance was declared at $p < 0.05$. All metabolomic data were analyzed using the MetaboAnalyst software [23]. Data normalization of metabolite concentration was done prior to statistical analysis (quantile normalization) to have a Gaussian distribution. In this study, we used log-transformed and auto scaled metabolite values. Multivariate analysis including principal component analysis (PCA), partial least squares-discriminant analysis (PLS-DA), and receiver–operator characteristic (ROC) curves were performed using MetaboAnalyst 3.0, following the protocol described previously by Dervishi et al. [17]. For the PLS -DA model, a permutation testing with 2000 random re-samplings was implemented to validate the reliability of the model and to determine the probability that the metabolites distinguishing the lame and CON groups are a result of chance. In the ROC curve analysis, a permutation testing was conducted for different timepoints with 1000 random re-samplings [23]. Additionally, in the PLS-DA model, a variable importance in the projection (VIP) plot was used to rank the metabolites based on their importance in discriminating the lame cows from CON cows. Metabolites with the highest VIP values are the most powerful group discriminators. Typically, VIP values >1 are significant and VIP values >2 are highly significant. For the biomarker analysis (i.e., ROC analysis), we picked the top five metabolites (VIP > 1.8) at −8 weeks, top four metabolites (VIP > 1.8) at −4 weeks, top three metabolites (VIP > 2.0) at the disease diagnosis w, top six metabolites (VIP > 1.6) at +4 weeks, and top three metabolites (VIP > 1.8) at +8 weeks, respectively. For exploration and visualization of the compounds network, we used Metscape plugin [24] in Cytoscape 3.5 [25]. The file containing the list of Kyoto Encyclopedia of Genes and Genomes (KEGG), IDs, fold change and *p*-values was loaded into Metscape.

## 3. Results

### 3.1. ¹NMR Urinary Analysis of Lame Cows

Results of this study showed that pre-lame, lame, and post-lame cows experienced alterations in the concentrations of multiple urinary metabolites. The alterations in the urine were present at the five timepoints in the study starting from −8 weeks prior to parturition and until +8 weeks after parturition. A total of 90 metabolites were identified and quantified.

### 3.2. Metabolite Alterations before Diagnosis of Lameness

Results showed a total of 27 metabolites that were altered in the urine of pre-lame cows at both −8 and −4 weeks prior to parturition ($p < 0.05$; Supplementary Tables S1 and S2). At −8 weeks, 19 metabolites (2-aminobutyrate, 3-hydroxy-3-methylglutarate, 3-indoxylsulfate, acetoacetate, adipate, Ala, ethanol, formate, methylmalonate, N,N-dimethylglycine, N-acetylglutamine, propylene glycol, salicylate, Ser, trimethylamine, Tyr, uracil, Val, and N-acetylglutamate) displayed a lower concentration in urine and eight metabolites (1,3-dimethylurate, ascorbate, hypoxanthine, Leu, Lys, O-phosphocholine, pantothenate, and xylose) were higher in the urine of pre-lameness cows when compared to healthy cows ($p < 0.05$). Uracil was the most decreased metabolite in the urine of pre-lameness cows, with a -9.36-fold change at −8 weeks prior to parturition.

At −4 weeks before parturition, the number of altered metabolites remained high. A total of 27 metabolites were altered, and among them five metabolites (formate, N,N-dimethylglycine, tyrosine, urea, and uracil) were excreted at a lower concentration in pre-lame cows compared to healthy counterparts and 22 metabolites (2-hydroxyvalerate, 3-aminoisobutyrate, acetylsalicylate, alloisoleucine, Asp, glucose, His, hypoxanthine, isocitrate, Lys, methanol, N-acetylaspartate, O-phosphocholine, pantothenate, Thr, tiglylglycine, Try, xylose, trans-aconitate, beta-alanine, pi-methylhistidine, t-methylhistidine) were excreted at a greater extent in pre-lame cows compared to healthy cows ($p < 0.05$). Urea was the most decreased metabolite in the urine of pre-lame cows, with a −6.18-fold change at −4 weeks before diagnosis of lameness. In addition, acetoacetate tended to be lower in the urine of lame cows ($p = 0.08$). Seven metabolites were consistently altered at −8 and −4 weeks prior to parturition ($p < 0.05$; Table 1). Four metabolites (formate, Tyr, uracil, and N,N-dimethilglycine (DMG)) were excreted to a lower extent, and three metabolites (pantothenate, Lys, and hypoxanthine) were excreted to a greater extend in pre-lame cows compared to the healthy cows.

**Table 1.** Concentrations of consistently altered urinary metabolites [mean (SD)] in healthy control (CON; *n* = 20) and lameness cows (*n* = 6) at −8 and −4 weeks prepartum.

| Metabolite (µM) | −8 Weeks Prior to Parturition | | | | −4 Weeks Prior to Parturition | | | |
|---|---|---|---|---|---|---|---|---|
| | **Lameness** | **Control** | ***p*-Value** | **Fold-Change** | **Lameness** | **Control** | ***p*-Value** | **Fold Change** |
| Formate | 24.63 (6.72) | 49.64 (29.68) | 0.01 | −2.02 | 27.88 (22.80) | 42.67 (24) | 0.03 | −1.53 |
| Tyrosine | 8.90 (2.76) | 15.40 (5.47) | 0.01 | −1.73 | 6.90 (4.36) | 13.06 (5.05) | 0.006 | −1.89 |
| Uracil | 15.66 (5.98) | 146 (99) | 0.001 | −9.36 | 29.96 (9.26) | 144 (97) | 0.01 | −4.82 |
| N,N-Dimethyl-glycine | 1.79 (0.79) | 4.07 (1.92) | 0.01 | −2.27 | 1.81 (0.95) | 4.01 (3.17) | 0.01 | −2.21 |
| Pantothenate | 13.12 (3.32) | 9.58 (3.47) | 0.03 | 1.37 | 12.89 (5.47) | 6.81 (3.04) | 0.001 | +1.89 |
| Hypoxanthine | 2.90 (0.92) | 1.77 (1.12) | 0.03 | 1.64 | 4.61 (3.49) | 1.73 (1.37) | 0.003 | +2.67 |
| Lysine | 30.59 (3.77) | 18.44 (7.77) | 0.001 | 1.66 | 32.28 (16.54) | 15.84 (6.79) | 0.001 | +2.04 |

Univariate analysis of data was performed using Wilcoxon–Mann–Whitney (rank sum) test. Statistical significance was declared at $p < 0.05$.

Multivariate analysis (PCA and PLS-DA) was used to analyze and reduce the data dimension. Multivariate analysis showed separation of pre-lame cows and CON ones based on concentrations of metabolites. Partial least squares-discriminant analysis (PLS-DA) had two separated clusters at −8 and −4 weeks prior to lameness diagnosis (Figure 1). The results of candidate biomarker analysis were performed using ROC analysis and they are summarized in Table 2. The top five and four important metabolites that separated the two groups at −8 and −4 weeks prior to parturition were

uracil, 2-amino- -butyrate, 3-hydroxy-3-methylglutarate, Val, and Tyr, as well as Lys, His, Tyr, and methanol, respectively. The cross-validation of the PLS-DA model using LOOCV method showed that Q2 > 0.7. Generally, a Q2 > 0.5 is considered "good", while a Q2 of 0.9 is considered outstanding.

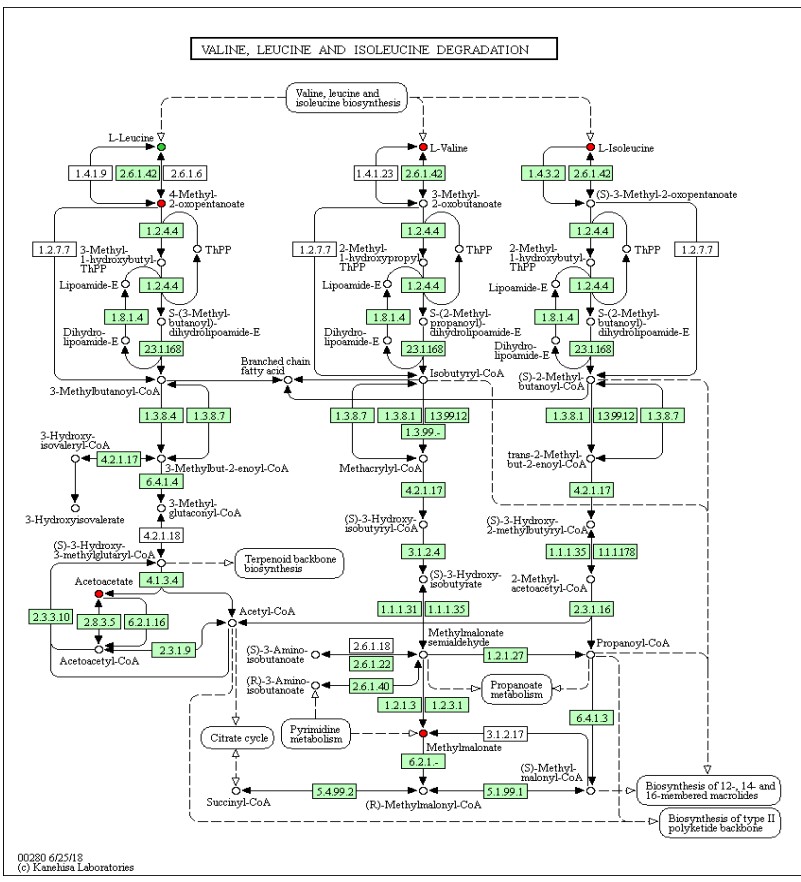

**Figure 1.** Visualization of Valine, Leucine, and Isoleucine degradation pathway. Red circles indicate lower concentration and green circles indicate greater concentration in lame cows.

**Table 2.** Biomarker profiles and the quality of the biomarker sets using receiver–operator characteristic curves.

| Metabolites | AUC | 95% CI | Adjusted *p*-Value |
|---|---|---|---|
| Uracil, 2-Aminobutyrate, 3-Hydroxy-3-methylglutarate, Valine and Tyrosine | −8 weeks before parturition 0.95 | 0.81–1 | 0.002 |
| Lysine, Histidine, Methanol, and Tyrosine | −4 weeks before parturition 1 | 1–1 | 0.001 |
| Tyrosine, Adipate and Glycerate | Week of LAM diagnosis 0.96 | 0.75–1 | 0.015 |
| N-Acetylaspartate, Glutamine, Imidazole, Pantothenate, beta-Alanine and Trimethylamine | +4 weeks after parturition 0.78 | 0.11–1 | NS |
| Hipurate, Pantothenate and 1,3-Dihydroxyacetone | +8 weeks after parturition 0.91 | 0.5–1 | NS |

AUC = area under the curve; CI = confidence interval; LAM = lameness, NS = not significant.

The combination of the top VIP score metabolites was significant at −8 and −4 weeks prior to parturition (empirical *p* < 0.05). The AUC for both curves were 0.955 (95% CI, 0.812-1) at −8 weeks and 1.0 (95% CI, 1 to 1) at −4 weeks prior to parturition, respectively (Table 2). These results indicate that the urinary metabolites identified can have a very strong predictive value for lameness in dairy cows.

The pathway analysis revealed that at −8 weeks prior to parturition, aminoacyl-tRNA biosynthesis, Val, Leu, and Ile degradation pathways, aminoacyl-tRNA biosynthesis, pentose and glucuronate interconversions, and pantothenate and CoA biosynthesis pathways were the top most enriched pathways. The significant metabolites (Val, Leu, and Ile degradation pathway) were used to identify and visualize metabolic pathways in KEGG (www.genome.jp/kegg/tool/map_pathway2.html). Changes

in Val, Leu, and Ile pathway are shown in Figure 1. Interestingly pre-lame cows excreted more Leu at −8 and −4 weeks prior to parturition and less acetoacetate and methylmalonate which might suggest that the amino acid Leu instead of being metabolized by the organism it is eliminated via urine (Figure 1).

The full list of altered urinary metabolites at −8 and −4 weeks prior to parturition are given at Tables S1 and S2 (Supplementary Materials). Additionally, the PCA, PLS-DA, ROC curve, and VIP analyses for weeks −8 and −4 prior to parturition are given in Figures S1 and S2 (Supplementary Materials), respectively.

### 3.3. Metabolite Alteration During Lameness Week

At the week of lameness diagnosis, a total of seven metabolites (2-hydroxyisobutyrate, 3-hydroxy-3-methylglutarate, 4-hydroxyphenylacetate, adipate, glycerate, Tyr, and Val) were found to be excreted at a lower concentration in cows diagnosed with lameness when compared with healthy group of cows (Table 3). Multivariate analysis showed the separation of cows diagnosed with lameness and healthy ones. Partial least squares-discriminant analysis had two separated clusters at −8 and −4 weeks before calving (Figure 2). The top five important metabolites that separated the two groups at the week of lameness diagnosis were Tyr, adipate, glycerate, 3-hydroxy-3-methylglutarate, and uracil. The combination of three top VIP score metabolites was significant (empirical $p < 0.05$).

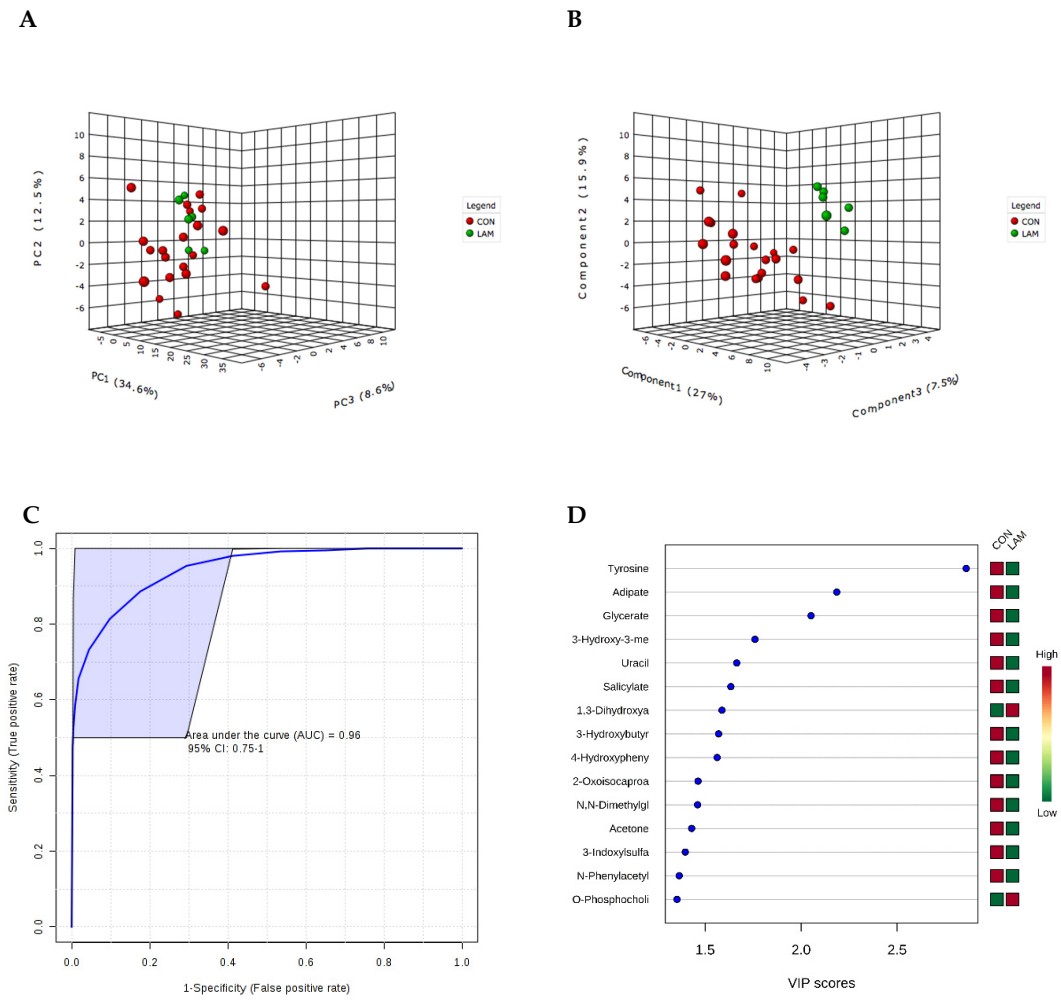

**Figure 2.** Visualization of (**A**) principal component analysis (PCA), and (**B**) partial least squares-discriminant analysis (PLS-DA) of 20 CON and 6 LAM (lame) cows at the week of lameness diagnosis, and (**C**) ROC curve and (**D**) variables ranked by variable importance in projection (VIP).

**Table 3.** Concentrations of altered urinary metabolites (mean (SD)) in lame cows (*n* = 6) and healthy controls (*n* = 20) at the week of diagnosis of lameness.

| Metabolite µM | Lameness | Control | *p*-Value | Fold Change |
|---|---|---|---|---|
| 2-Hydroxyisobutyrate | 4.46 (0.53) | 5.74 (2.08) | 0.02 | −1.29 |
| 3-Hydroxy-3-methylglutarate | 7.16 (2.05) | 12.52 (6.31) | 0.02 | −1.75 |
| 4-Hydroxyphenylacetate | 7.54 (3.60) | 18.98 (13.15) | 0.001 | −2.52 |
| Adipate | 60.11 (14.01) | 96.98 (33.54) | 0.002 | −1.61 |
| Glycerate | 42.0 (21.44) | 85.14 (55.35) | 0.01 | −2.03 |
| Tyrosine | 8.54 (3.50) | 24.44 (10.66) | 0.0001 | −2.86 |
| Uracil | 48.78 (14.23) | 194.56 (153.12) | 0.0004 | −3.99 |
| Urea | 1707 (464.55) | 7893,42 (6576,60) | 0.0005 | −4.62 |
| Valine | 8.21 (2.06) | 11.86 (5.74) | 0.02 | −1.44 |

Univariate analysis of data was performed using Wilcoxon–Mann–Whitney (rank sum) test. Statistical significance was declared at *p* < 0.05.

### 3.4. Metabolite Alterations after Diagnosis of Lameness

Cows that were affected by lameness continued to have alterations in urine metabolites at +4 and +8 weeks after parturition in comparison with control cows. More specifically, cows with lameness excreted greater quantities of glutamine, imidazole, N-acetylaspartate, and pantothenate via urine when compared with control cows (Table 4). At +8 weeks after parturition, cows with lameness excreted greater 1,3-dihydroxyacetone, galactose, hippurate and pantothenate compared with healthy cows (Table 3).

**Table 4.** Concentrations of altered urinary metabolites (mean (SD)) in lame cows (*n* = 6) and healthy control (*n* = 6) at +4 and +8 weeks after diagnosis of lameness.

| **+4 Weeks after Parturition** | | | | |
|---|---|---|---|---|
| Metabolite µM | Lameness | Control | *p*-Value | Fold Change |
| Glutamine | 59.73 (40.63 | 25.97 (11.27) | 0.04 | +2.3 |
| Imidazole | 68.80 (46.40) | 30.72 (5.87) | 0.02 | +2.24 |
| N-Acetylaspartate | 36.02 (19.12) | 11.27 (3.83) | 0.02 | +3.2 |
| Pantothenate | 29.28 (11.75) | 16.65 (10.14) | 0.04 | +1.76 |
| **+8 weeks after parturition** | | | | |
| 1,3-Dihydroxyacetone | 16.76 (6.76) | 6.16 (4.14) | 0.008 | +2.72 |
| Galactose | 21.48 (6.49) | 10.63 (9.23) | 0.04 | +2.02 |
| Hippurate | 166.94 (67.97) | 41.67 (31.89) | 0.002 | +4.01 |
| Pantothenate | 25.49 (12.33) | 12.33 (8.92) | 0.04 | +2.07 |

Univariate analysis of data was performed using Wilcoxon–Mann–Whitney (rank sum) test. Statistical significance was declared at *p* < 0.05.

Interestingly at +4 and +8 weeks after parturition, lame cows excreted greater amounts of pantothenate via urine than the healthy controls. PLS-DA analysis at +4 weeks after calving revealed that N-acetylaspartate, glutamine, imidazole, pantothenate, beta-alanine and trimethylamine had the greatest VIP score at +4 weeks after calving. In addition, hipurate, pantothenate 1,3-dihydroxyacetone, galactose, and tyrosine had the greatest VIP score at +8 weeks after calving. Regarding biomarker analysis, the combination of those metabolites at +4 and +8 weeks after calving did not produce significant results (empirical *p* > 0.05).

The PCA, PLS-DA, ROC curve, and VIP analyses for weeks +4 and +8 after parturition are given in Figures S3 and S4 (Supplementary Materials), respectively.

## 4. Discussion

Lameness in dairy cows is a multifactorial disorder, which progresses gradually and has not been characterized from the metabolic prospective. To better understand the full pathobiology of lameness,

90 urinary metabolites were identified and quantified in 20 healthy CON and six lame cows using an $^1$H-NMR-targeted metabolomics approach. The most important finding of the study is that multiple urinary metabolites were identified as altered in pre-lame, lame, and post-lame cows that could be used as potential monitoring or screening biomarkers for lameness. In particular, four metabolites including uracil, formate, N,N-dimethylglycine (DMG), and Tyr were consistently lower, whereas another four metabolites were greater (Lys, pantothenate, hypoxanthine, and xylose) in the urine of pre-lame cows (i.e., at −8 and −4 weeks prepartum), respectively. Moreover, similar alterations were also observed at lameness week as well as during the postpartum period.

With regards to the potential cause of lameness in cows in our study, it is speculated that lameness is related to nutritional factors and, more specifically, to feeding high-grain diets immediately after calving. Cows were fed high-grain diets (at 45% of DMI) to support energy requirements for lactation and were diagnosed as lame 2–3 weeks after parturition. Previously, we reported that feeding grain diets at 45% of DMI is associated with a 13.5-fold increase in the amount of endotoxin in the rumen fluid and with a systemic inflammatory state [26]. The most important metabolic changes are discussed in detail below.

Formate is an essential intermediary metabolite in folate-mediated one-carbon (1C) metabolism. Formate is generated in mitochondria during the conversion of Ser to Gly. Interestingly, Ser was lower in the urine of pre-lame cows (at −8 weeks prepartum). This is an indication that lower formate in the urine of pre-lame cows might be related to the inhibition of serine catabolism during the prepartum period. Dimethylglycine, which is produced from choline catabolism, is another important source of 1C units. One-carbon units support multiple physiological processes including biosynthesis of nucleotides (purines and thymidine), amino acid homeostasis (Gly, Ser, and Meth), methylation processes (epigenetic maintenance), and redox defense mechanisms [27]. Serine has also been shown to be the major source of 1C units in proliferating lymphocytes [28]. It should be noted that, during mitochondrial respiratory dysfunction, the cellular 1C metabolism is altered and is associated with lowered formate production from serine [29]. The latter authors also showed that dysfunctional mitochondria are less able to utilize serine to produce formate, which is a precursor for the DNA building blocks purine and thymidine. If cells do not have enough serine to compensate for this inefficiency, they cannot produce some of the essential compounds necessary for the synthesis of DNA and other important compounds in the cells, especially immune cells. Therefore, based on the data obtained, it can be speculated that, in pre-lame and lame cows, the cellular 1C metabolism is altered and is associated with lowered formate production from serine [29].

Urinary Tyr also was found to be lower in pre-lame and lame cows, but not in post-lame cows at +4 and +8 weeks postpartum. In human medicine there has been reports indicating that during inflammatory states conversion of phenylalanine to tyrosine is impaired and their ratio is elevated. Our results agree with these reports, given that both pre-lame and lame cows had higher Phe/Tyr ratio in the serum at −4 weeks (1.44 vs. 1.23) prepartum and at +4 (1.49 vs. 0.50) and +8 weeks (1.66 vs. 0.86) postpartum.

Moreover, the Phen/Tyr ratio in the urine of those cows was even higher at both −8 (3.67 vs. 2.75) and −4 weeks (5.17 vs. 2.63) prepartum, as well as during the disease week (3.76 vs. 1.95). It has been reported that the diminished conversion of phenylalanine to tyrosine by phenylalanine hydroxylase (PAH) may be due to an increased output of reactive oxygen species (ROS) produced by macrophages upon activation [30]. The low functional activity of PAH also might be related to activation of the Th1-type immune responses and/or a deficiency of tetrahydrobiopterin (BH$_4$) [31,32]. Indeed, in a companion article we reported that pre-lame cows experienced activation of innate immune responses characterized by elevated levels of pro-inflammatory cytokines and acute phase proteins including interleukin-6 (IL-6) and tumor necrosis factor (TNF), as well as acute phase proteins haptoglobin (Hp) and and serum amyloid A (SAA) in the serum at −4 weeks prepartum and during lameness diagnosis [12]. Elevated proinflammatory cytokines and acute phase proteins in pre-lame and lame cows suggest the presence of a chronic low-grade inflammatory state in those cows associated with

alterations in the Phe/Tyr ratio. Similar associations between inflammatory conditions like trauma or sepsis and high Phe/Tyr ratios have been described previously in human subjects [33].

Other significant findings of this study were lower urinary uracil in pre-lame cows at both −8 and −4 weeks prepartum and elevated pantothenate during all five timepoints included in the study. In fact, uracil was ranked as the top metabolite, with the highest VIP score, in the predictive biomarker model at −8 weeks prepartum. Uracil is one of the key pyrimidine metabolites, which are essential components of nucleic acids. Uracil is a precursor to alanine and pantothenate, known as vitamin B5, which is a cofactor of coenzyme-A [34]. Given that uracil was lower and pantothenate higher in the urine of pre-lame and lame cows, this suggests that uracil might have been used for the synthesis of pantothenate. Pantothenic acid plays pivotal roles in the metabolism of carbohydrates, lipid synthesis (as part of the acyl carrier protein), and proteins in ruminants [35]. Increased production of the free form of pantothenate and its excessive leakage in the urine of pre- and post-lame cows can have negative implications on the overall metabolic performance of the cows.

Data also showed a high excretion of urinary xylose in pre-lame cows vs. their healthy counterparts. Xylose is a five-carbon monosaccharide found in plants, which is not degraded in the gastro-intestinal tract, and it is absorbed unchanged [36]. It is obvious that diet is not the reason for the difference in xylose excretion in the urine between the two groups of cows because the ration offered to both groups prior to calving was the same. This prompted us to look for other potential reasons for the high urinary xylose. It is important to note that xylose is not degraded in the body and it is not used for the generation of energy; however, xylose is a structural part of the glycocalyx that covers the vascular side of the endothelial cells [37]. Glycocalyx is a sugar-rich layer located at the luminal part of the endothelial cells throughout the vasculature. During inflammatory conditions, as is the case with our pre-lame cows, glycocalyx fragments are shed under the influence of proinflammatory cytokines and mediated by metalloproteinase-9 [38]. Glycocalyx fragments act as damage-associated molecular pattern (DAMP) ligands, binding to toll-like receptor-4 and increasing proinflammatory cytokine production [39]. It has been demonstrated that, during experimental endotoxemia or sepsis, there is degradation in glycocalyx associated with microcircular dysfunction, as happens in the microcirculation of the hoof area during laminitis, well-studied in horses [40–42]. Therefore, it is possible that a chronic low-grade inflammatory state prior to clinical lameness might play a role in injuring the vascular endothelial cells and initiation of the pathological process. Moreover, whether xylose, which is part of the glycocalyx, might be used in the future as an early biomarker of vascular endothelial cell damage in pre-lame dairy cows remains to be explored.

Urea was another important urinary metabolite that differentiated the two groups of cows with pre-lame and lame cows having lower urinary concentrations compared with the healthy CON at −4 weeks prepartum, as well as at lameness week. Indeed, urea concentrations in the urine of pre-lame (−8 and −4 weeks) and lame cows (lameness week) were lowered by −3.93-, −6.18-, and −4.62-fold compared with healthy CON cows. Synthesis of urea occurs in the liver and it takes two molecules of bicarbonate ($HCO_3^-$) and two molecules of ammonium ($NH_{4+}$) to synthesize one molecule of urea [43,44]. Both bicarbonate and ammonium are released from the degradation of amino acids in the liver. It should be noted that pre-lame and lame cows were in a state of chronic low-grade lactic acidosis because concentrations of lactate in blood at −8 and −4 weeks prepartum, as well as at lameness week, were higher in those cows [12]. It is known that, during metabolic acidosis, the synthesis of urea from the liver slows down to save bicarbonate molecules which are important to neutralize the acidity of blood caused by lactic acidemia [43]. Therefore, lower urinary urea in our cows might be a compensatory response of pre-lame and lame cows to maintain the acid-base balance in the body.

Urinary Lys also was greater in pre-lame cows at −8 and −4 weeks prepartum. It is known that Lys is a ketogenic amino acid that is catabolized into acetoacetate and β-hydroxybutyrate (BHB) [45]. However, concentrations of BHB were not greater in the serum or urine of pre-lame cows [12]. Therefore, it is possible that Lys was used for other purposes by the host and not for the generation of ketone

bodies. It can be speculated that lysine might have been used for the mounting of an immune response by pre-lame cows. As indicated previously, pre-lame cows showed greater proinflammatory cytokines (i.e., IL-6) and acute phase proteins (Hp and SAA) in the serum, starting from −4 weeks prepartum, and during lameness week [12]. Iseri and Klasing [46] demonstrated that, during an acute phase response to an intravenous challenge with *Escherichia coli*, the amount of lysine to B cells almost doubled 5–7 days post challenge and remained high for 21 days. Moreover, the content of lysine in spleen B cells increased by 10-fold. Therefore, it is possible that lysine might have been released in large amounts from skeletal muscles to provide immune cells with the necessary resources to mount an immune response to the potential bacterial or endotoxin insults during the dry-off period and during lameness week.

It was interesting that the urinary metabolite differentiation between the two groups of cows continued to be present, even during the post-lameness week at +4 and +8 weeks postpartum. Thus, at +4 weeks postpartum, N-acetylaspartate, Glu, imidazole, pantothenate, and beta-alanine were greater in post-lame cows. Intriguingly, even at +8 weeks postpartum, several urinary metabolites were still altered in post-lame cows, including hippurate, pantothenate, and 1,3-dihydroxyacetate. Those metabolites were classified in the VIP score as the most important ones distinguishing the two groups of cows. Differences between the two groups might partially explain the impact that lameness has on the reproductive performance and milk production of dairy cows postpartum.

Caution is advised with the interpretation of these data, given that the number of lame cows is low. Therefore, the data should be considered as preliminary, needing further validation in the future to confirm that alterations identified are typical of lameness.

## 5. Conclusions

Overall, the results of this study showed typical urinary metabotypes during the pre-lame stages, during lameness week, as well as several weeks after lameness incident. The most important and consistent alterations were found during the pre-lameness period as well as during the lameness week. Thus, urinary uracil, formate, N,N-dimethylglycine (DMG), and Tyr were consistently lower, and Lys, pantothenate, hypoxanthine, and xylose were greater in pre-lame cows at −8 and −4 weeks prepartum. Given the findings of this study, it can be concluded that it is possible that urinary metabotyping can be used to screen dairy cows for susceptibility to lameness 6–10 weeks prior to lameness week. Data also suggested that pre-lame and lame cows might have mitochondrial dysfunctional issues in the generation of 1C unit providers, including formate and DMG as well as the inhibition of urea synthesis to oversee metabolic acidosis. Finally, elevated urinary xylose in pre-lame and lame cows suggests potential damage to the glycocalyx layer that covers the vascular endothelial cells, that might be a very early stage of the impact of a chronic low-grade inflammation on microciculatory dysfunction, commonly observed in lame animals. The potential cause of lameness might be the susceptibility of some of the cows (the lame ones) to the high-grain diet offered immediately after parturition. Finally, the results of this study are intriguing, however, it should be pointed out that these results are preliminary and, given the low number of replicates, these data need to be validated in the future in a larger cohort of animals.

**Supplementary Materials:** The following are available online at http://www.mdpi.com/2624-862X/1/1/2/s1, Figure S1. Visualization of (A) principal component analysis (PCA), and (B) partial least squares-discriminant analysis (PLS-DA) of 20 CON and 6 LAM (lame) cows −8 weeks prior to parturition, and (C) ROC curve and (D) variables ranked by variable importance in projection (VIP), Figure S2. Visualization of (A) principal component analysis (PCA), and (B) partial least squares-discriminant analysis (PLS-DA) of 20 CON and 6 LAM (lame) cows −4 weeks prior to parturition, and (C) ROC curve and (D) variables ranked by variable importance in projection (VIP), Figure S3. Visualization of (A) principal component analysis (PCA), and (B) partial least squares-discriminant analysis (PLS-DA) of 20 CON and 6 LAM (lame) cows +4 weeks after parturition, and (C) ROC curve and (D) variables ranked by variable importance in projection (VIP), Figure S4. Visualization of (A) principal component analysis (PCA), and (B) partial least squares-discriminant analysis (PLS-DA) of 20 CON and 6 LAM (lame) cows +8 weeks after parturition, and (C) ROC curve and (D) variables ranked by variable importance in projection (VIP), Table S1. Concentrations of altered urine metabolites [mean (SD)] in lame and

healthy control cows at −8 weeks before parturition., Table S2. Concentrations of altered urine metabolites [mean (SD)] in lame and healthy control cows at −4 weeks before parturition.

**Author Contributions:** Conceptualization: B.N.A. and D.S.W.; Animal work: G.Z. (Guanshi Zhang) and E.D.; Metabolomics analysis: G.Z. (Guanshi Zhang) and R.M.; Data processing and statistics: G.Z. (Guanshi Zhang), E.D.; Writing—original draft: E.D. and G.Z. (Guanshi Zhang); Writing—Reviewing and Editing: B.N.A., E.D., G.Z. (Guanshi Zhang), G.Z. (Grzegorz Zwierzchowski), R.M. and D.S.W. All authors have read and agreed to the published version of the manuscript.

**Funding:** This research was funded by Genome Alberta (Calgary, AB, Canada) and ALMA (Alberta Livestock and Meat Agency Ltd., Edmonton, AB, Canada), grant number AARI2008A100R.

**Acknowledgments:** We thank Genome Alberta (Calgary, AB, Canada) and ALMA (Alberta Livestock and Meat Agency Ltd., Edmonton, AB, Canada) for financial support of the project. We also thank the full or partial contribution of D. Hailemariam, S.A. Goldansaz, Q. Deng, and J.F. Odhiambo in collection of samples from the cows.

**Conflicts of Interest:** The authors declare no conflict of interest.

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
