# Peer review of "Urinary Metabolomics around Parturition Identifies Metabolite Alterations in Dairy Cows Affected Postpartum by Lameness: Preliminary Study"

_2624-862X, doi:10.3390/dairy1010002_

Round 1
Reviewer 1 Report
Using urinary metabolites as an indicator of lameness in dairy cattle is a novel idea, and the authors are applauded for the work. There a number of points that are missing from the narrative however, I have outlined them below:
The authors mention “animal welfare” in the simple summary – however they do not really follow up on it anywhere in the rest of the document. Presumably the animal welfare statement refers to the condition being painful? Currently the introduction starts with a summary of costs – however, the costs are actually just an outcome of the main issue – which is the pain experienced by the cow, which then results in a change in behaviour, which then ultimately results in things like decreased feed intake, sickness behaviour, lower milk production, reduced reproductive function, etc. Please precede the costs/production paragraph with a discussion about what lameness is, and why it mattersTo follow the above paragraph – a consideration for what causes lameness is extremely important. The metabolic response of a cow experiencing a sole ulcer is going to be very different from one experiencing acidosis, or one with digital dermatitis (etc). This will be very important for the other additions to the paper… which is to discuss the causes of lameness in the 6 cows. I understand this might have been discussed already in addition to the treatments in a previous paper – however it is instrumental for the narrative here and must be included.
The discussion requires a limitations section. First off, while the larger study design is a case control, this subset is really an observational study – and therefore that needs to be mentioned. Second, there are just 6 cows. Yes, the authors found significant metabolites – however, the current narrative reads as though these should be used to identify potentially lame cows. There is no indication how representative of the general population of cows these 6 cows actually are – (nor should they be considered a representative sample either!)
Please have an native English speaker review – as there are numerous (small) grammatical errors that if fixed would help in the flow of the document.
Ln38 – post-partum (to remain consistent in wording)
Ln 44. Keywords should not include words in the title.
Ln 60-61 – difficult to follow
Ln 74 and elsewhere – ‘during lameness event’ – is this during THE lameness event? But actually, shouldn’t it be really referring to when lameness was first identified? (animals are not lame for an event, rather a duration of time; the event is when they are diagnosed)
Ln 82 – please provide the name of the animal ethics committee that reviewed this study and the approval #
Ln 89 – this section is a hard to follow. My suggestion would be to not mention the number of animals in the larger study – just refer to it, and then focus on the numbers in this observation… which is a bit mixed up. Ln 95 suggests there are 6 healthy cows, but elsewhere there are 20 healthy cows and 6 lame ones. Please clarify.
Ln 113 – as previously – causes need to be included.
Ln114 – how were urine samples actually taken from the cows? (details of procedure, equipment etc.)
Ln154-157. Please provide more details regarding the specific models. It would be very difficult to replicate the analysis right now. In particular, I’d like to see the number of models run, and the n that went into each model. How was the PCA actually built – there are a lot of variables and a small number of cows (traditionally – a PCA cannot handle more variables than there are subjects!)
Ln 168. Remove ‘in cows involved in the …’
Ln 192 – 193. Refer to these in methods.
Figure 1. I’m not following how the PCA was able to be run on so many variables with so few subjects. Please clarify in methodology.
Table 1. provide n in description
Table 1,3,4. why SD and not SE?
Table 1. lame (not lameness), health (not control)
Table 1. include ‘fold change’ in methods
Table 1. There are 6 lame cows and 20 healthy … how come the variability is so much higher in the healthy cows? (it’s very rare to have a situation where less animals results in such a much lower variability than more animals!)
Figure 2. missing?
Ln247. Produce (not produced)
Ln248. Define significance in the stats section and don’t mention it again in the results (if you report a model is different, than it is presumed statistically different)
Ln257-259. Repetitive delete
Ln267. This sentence suggests that the current work is helping to elaborate on better understanding lameness in dairy cows. I’d suggest that due to its observational nature and small sample size, this study actually presents preliminary evidence that metabolites MIGHT be useful. Please amend.
Discussion – generally, the authors do discuss each of the metabolites of interest and present backing literature… however what is missing is the underlying ‘why do these variables come up significant?’
This is where the authors are failed by their lack of discussion on causes of lameness. Please insert a table in the results documenting each cow’s cause of lameness… and work that into the narrative. This is important – for instance, on ln 303 the authors indicate that high Phe/Tyr ratios are noted in humans suffering from trauma or sepsis… however, how many of the 6 cows in the present study had either of these? (is a sole ulcer the same as laminitis? No!)
Ln383. Would all lame animals have chronic low-grade inflammation? (again, this comes back to the small sample size)
Ln383-385. Can I encourage the authors to conclude with something other than the typical “more research is needed” statement?
Reviewer 2 Report
Abstract: Conclusion- kind of a bold last statement to make with such a small sample size.
Define VIP in the abstract- most are unfamiliar with this term.
Tables 1 and 2: it seems strange that some of the ROC values are so high, even with such a small sample size. I think reporting sensitivity and specificity of the ‘optimal cut-off’ is needed in order to determine the actual predictive value of these biomarkers. It is also strange that some of the 95% CI were 1 to 1 with no variation?
L20: reword “so that to allow interventions..” sounds strange
L26: Analyze instead of “search”
L50: what does “with high score of 4” mean?
L77: “progression” instead of progress.
L95: were these 6 healthy cows selected from the original 20 healthy control cows?
L106-112: Who was the scoring done by? One or more people? What is the reliability of this scoring system used?
L257: “characterized” not “characterize”
Figure 1 doesn’t really illustrate the sentence that it is attached to. I think adding a few figures (can do supplementary if not enough room) on how the PLS-DA plots and the PCA plots looked would really strengthen the analysis. There is a reference to a figure 2, but I do not see a figure 2 in the manuscript.
L266-267: This last sentence isn’t clear what you mean by “elaborated to a new understanding”
L278: Do you mean latter authors?
Tables: in the footnotes mention what statistical test was used for comparisons.
S1 and S2 tables: it isn’t -8 weeks before lameness diagnosis, correct? 8 weeks before parturition?
I’ve noticed throughout the tables, there is inconsistently in saying “-8 and -4 weeks prepartum” versus “-8 and -4 weeks prior to lameness diagnosis.” Make sure these are correct and consistent.
Add sample sizes on tables.
Table 2: What does “No” mean under adjusted p-value?
Great discussion of what possible biological mechanisms link these biomarkers to lameness. I think the discussion is missing a section on limitations of the current analysis and what future studies are needed. I think a big limitation is the small sample size of this study and that larger studies are needed to be able to determine the optimal thresholds of these biomarkers.
